# Geoethics, a Branding for Sustainable Practices

**Martin Bohle** [1,2,*] and **Eduardo Marone** [2,3]

1   Ronin Institute for Independent Scholarship, Montclair, NJ 07043, USA
2   International Association for Promoting Geoethics (IAPG), 00143 Rome, Italy
3   Centre for Marine Studies, International Ocean Institute Training Center for Latin America and the Caribbean (IOITCLAC), Federal University of Paraná (CEM/UFPR), Pontal do Paraná 83255-976, Brazil; edmarone@gmail.com
*   Correspondence: martin.bohle@ronininstitute.org

**Abstract:** In struggles for cultural leadership, advocating a paradigm helps to disseminate, for example, a style of life, thinking, or common practices. Promoting a practice, that is, branding it, includes the use of a simple name or symbol (semiotic sign). Within geosciences, the label "geoethics" refers to a school of thought that uses established philosophical concepts to promote responsible professional practices. The outcomes that are available aggregate to a more general paradigm that calls for geocentric human practices. The label geoethics also sounds like a brand for those practices. As analysis shows, the notion of geoethics is not univocal. At first sight, that feature hinders using it as a brand for geocentric practices. However, the successful branding of the concept of sustainability, as a scientific and public paradigm, indicates the opposite. Although the notion of sustainability aggregates various concepts and is not univocal, it illustrates what cultural leadership can be achieved when a concept, paradigm, and brand use the same semiotic sign (name). Therefore, it is suggested that the school of thought, Geoethics, with its dedicated reference to the specific societal use of geosciences, should also be used as a brand: geoethics, the general application of geoethical thinking to promote geocentric human practices.

**Keywords:** geoethics; geosciences; sustainability

## 1. Introduction

"Sustainability" is a concept, a notion, and a word that has also acquired worldwide recognition as a "brand". "Branding" is the *"practice of creating a name, symbol or design that identifies and differentiates a product from other products"* [1]. That simple, commercial concept illustrates the semiotic importance as to how signs and symbols create meanings and, in such a sense, branding the word (symbol) "geoethics" means to give an identity that differentiates this sign from any other.

Scholarly definitions of sustainability are many [2,3], and the implications of the concept of sustainability for human ethical practice are evident [4–7]. The notion of Geoethics labels a specific school of thought within geosciences that considers sustainability as a value [8] (p. 48). As a school of thought, Geoethics may be situated at the intersection of different ethical frameworks (environmental, sustainability, engineering, or professional ethics) [9] (pp. 4, 166). The word geoethics is not recognized as a notion that describes a branding of the set of concepts and practices, which Geoethics as philosophical thinking within geosciences aggregates. In turn, the notion of geosciences refers to *"a range of applied and fundamental research fields, as well as related engineering disciplines and commercial undertakings. Together, they address the functioning of the Earth, the intersections of Earth and human systems, as well as the extraction and use of (abiotic) natural resources"* [9] (p. 171).

In this essay, we describe why the brand geoethics is meaningful; for example, because of the intrinsic relation between sustainability and geosciences [8,10,11]. Studies in geoethics are about *"research and reflection on the values that underpin appropriate behaviors*

*and practices, wherever human activities interact with the Earth system"* [12] (pp. 4–5) [13]. However, considering the ambition that is expressed in this phrase, perceiving geoethics as a competitor of, for example, spiritual concepts such as Gaian Ethics [14] is a mistake. The sciences of the abiotic compartments of Earth are at the origin of geoethics, which render it, at least on first sight, anthropocentric and a-historic (because of the manner in which it refers to the activities of people).

The Earth, as a human place, exhibits interwoven physical, cultural, and technological dimensions [15–18]. Nature belongs to that place and is tightly interrelated on any dimension of the human place. Lasting, sustainable interrelations are the essence of the human condition [19]. Within that human place, that is, within the human niche [20] or planetary social-ecological system [21], the ethics of human practices are about what ought to be done [22,23]. They set pivotal paradigms for human practices. Various ethical frameworks are available; that is, there are various schools of environmental or sustainability ethics [24,25], including some that use the label geoethics [26].

A paradigm encapsulates a generic view or set of ideas about how to perceive something. Branding the geoethical paradigm is not only about the rational or epistemic backgrounds of Geoethics. It goes beyond that, offering a sense-making of the paradigm by also appealing to emotional roots, or as Mike Begon [27] said, *"we should adopt, and repeat, emotionally appealing catch phrases"* (p. 394). Branding is a way of emotionally appealing to other people, particularly outside academia [28]. Thus, the branding exercise acquires an unquestionable semiotic relevance, giving the sign of geoethics rational and emotionally appealing meanings.

Branding serves in a cultural-political struggle to shape meaning and content. The naming of a paradigm does function as a semiotic sign, which triggers the affective societal contexts of the ideas that the paradigm encapsulates [29]. Branding through a semiotic sign is about affective sense-making of insights and practices at the level of social groups [30,31]. Affective anchorage in social and cultural lives is a precondition of a brand and renders it operational. Therefore, appropriate societal contexts of the scientific content must be given to brand a (scientific) paradigm. The brand sustainability may serve as an example, for which Gro Harlem Brundtland recently summarized the societal contexts:

> *that we will only secure a prosperous, peaceful and liveable planet if we harness economic growth and development to social solidarity across and between generations . . . Today, faced with the imperative of tackling climate change and responding to radical, fast-paced shifts in global technology, consumption and population patterns, there is growing consensus that sustainable development is the only way that we can avert environmental and social disaster . . . I have always believed that the development of science itself must be informed by humane values, and its awesome power must be applied in ways that respect human rights and share the benefits of progress in an equal and just fashion.* [32] (pp. xv–xvi)

Geoethics is discernible within the geosciences as a school of thought and has been for a bit more than a decade. Sustainability, as a scientific school of thought, is much older, and it exists as an accepted political concept. However, sustainability is not the hegemonic societal concept of the contemporary globalized society, although it is part of the cultural mainstream. Compared to it, geoethics is more like an ambition, although the notion of geoethics sounds like a branding. The meaning of the term seems evident. It sounds like a call to human agents to have an Earth-centric perspective and practices.

Geoethical practice takes its strength from the fact that it is anchored in the geosciences, the sciences of the functioning of a habitable planet [33]. As quoted above, works in geoethics are about *"research and reflection on the values that underpin appropriate behaviors and practices, wherever human activities interact with the Earth system"* [12] (pp. 4–5) [13]. That phrase also coins the paradigm geoethics.

Whether geoethical practice has the potential to be branded beyond the professional geosciences is the research question of this essay. Assessing it requires going beyond

concerns for the environment, economic growth, and social solidarity to consider the challenge of anthropogenic global change [34]; hence, the defiant Earth [35].

Following this introduction, the essay is structured in three sections: first, the contexts for a paradigm of geoethics are described; second, the branding potential is discussed; and third, it is concluded how to situate geoethics as a brand.

## 2. Contexts for a Paradigm "Geoethics"

This section is subdivided into five parts: a brief description of geoethics (for a comprehensive description, see [8]); an outline of the sociohistorical context of geosciences; a sketch of the affective connotations of the central geoethics paradigm; a description of the ambiguous use of the term geoethics in the scientific literature; and a reference to the use of geoethics in the public sphere.

### 2.1. Sketching [G][g]eoethics

The concepts that underpin geoethical thinking (Geoethics) draw only on Western cultures, as the semantic roots of the (anglicized) notion of geoethics encapsulate nicely. Specifically, Geoethics draws on Kantian models that put the human agency at the center [36]. No reference is made to cultural models of other parts of the world (e.g., non-Western models, such as the African Ubuntu culture or Buddhist, Chinese, Indian, or Islamic cultures).

Initially, Geoethics was about deontological concerns, that is, how geo-professionals situate themselves in professional contexts [37–39]. Geoethics is designed as a virtue-ethics with the responsible and (geoscience) knowledgeable individual as a central (Kantian) tenet. It has been proposed to situate geoethics at the intersection of sustainability ethics, environmental ethics, and professional ethics [40]. That arrangement is justified because geoethics is supporting professionals who apply geoscience expertise.

Studying the concepts that underpin ethical practices in geosciences led to insights into how Geoethics may support any citizen when intervening in the Earth system [23]. Explicitly studying further ethical practices for times of global anthropogenic change [34] led the authors of this essay to consider Geoethics as a reflection on sound governance practices to navigate the human niche [36]. Hence, studying ethical practices in geosciences indicates that Geoethics may be of more diverse use than initially thought, thus scaling up a simple deontological level to broader-applied ethics inquiries going beyond geoscientific professional practices.

Geoethics uses normative settings of an intermediate level, such as calls to act in a responsibility-focused way. It does so without explicitly referring to specific ethical frameworks (e.g., Rawlsian ethics of justice [7]) that would prescribe what ought to happen. Instead, geoethics exhibits a relativism (that is a pluralism of ethical frameworks) constrained by scientific knowledge [41] to favor *"context-dependent in space and time and ethically sound choices . . . [and] a strong awareness of the technical, environmental, economic, cultural and political limits existing in different socio-ecological contexts"* [13]. Hence, Geoethics exhibits a systemic relativism; namely, it leaves the human agent the choice whether to use a specific ethical framework. Consequently, what is understood in geoethics as sound ethical practice may alter with the ethical framework that is used by the human agent [42] and, mutatis mutandis, what is perceived as a sound geoethical practice varies depending on whether Utilitarian concepts [43], Prioritarianism [44], or a Rawlsian variant of ethics of justice [7] are applied. The benefit of such a "relativism by design" is that a diversity of cultures can be accommodated if two central tenets are observed, namely "individual accountability" and "scientific knowledge-base". The first tenet implies a substantial Kantian overhead. The second tenet implies deep roots in the history of the European Enlightenment. Therefore, the "relativism by design" of Geoethics is less a moral relativism but more the outcome of being an epistemic-moral hybrid [45].

## 2.2. The Sociohistorical Context of Geosciences

Geoethics is a product of Western culture. Hence, it should be put into the associated sociohistorical context. For example, the wording of the definition of geoethics, namely "human activities interact with the Earth system", conveys a basic concept of European origin, namely, the debatable "dichotomic stance" of Nature versus World [46–50].

During the last 500 years, post-medieval European cultural models paved a hegemonic global development path [51], including at its core the capitalist political economy and its related paradigms [52]. Overthrowing the economy, culture, and science of medieval societies, the Enlightenment offered new paradigms [53,54]. As a result, the foundations of what is today branded as Western culture were cast. The label, Western culture, habitually refers to the hegemonic Anglo-Saxon variant. Nevertheless, the global outreach of Dutch, French, German, Italian, Portuguese, Russian, and Spanish powers (alphabethic order) between the 15th and 20th centuries are intrinsic to the global expansion of European cultural models.

Together, European cultural models mobilized gigantic physical, mental, and economic resources [55,56] and also engaged (early on) with the sciences of the Earth [57,58]. A vast sociopolitical development was the consequence (e.g. the ability to sustain a much bigger human population): early global change occurred [59], massive distortions including slavery and colonialism happened [60,61], and recently, social-ecological problems at the planetary level emerged [62–64]. None of these developments would have been possible without the application of geoscientific know-how, irrespective of whether individual, collective, or institutional (human) agents have been aware of it or not. Along with the sociohistorical development of the second half of the last millennium, the perception of World and Nature varied, taking Purdy's account of North American perceptions [43] as one example of many.

Today, ample evidence has accumulated that the given societal practices of globalized capitalist society may not sustain the current physical stage of the Earth system [65] that geology calls the "Holocene". Subsequently, a phase shift of the Earth system seems likely. An alternative stage, tentatively named "Anthropocene", may fall into place [66]. Climate change is the best-known example of anthropogenic global change. However, the anthropogenic change of the global nitrogen cycle [67], starting with the development of industrial nitrogen fixation before WWI [68], is illustrative of anthropogenic climate change. Changing the global water cycle through damming is another example [69].

Notwithstanding whether the epoch "Anthropocene" is added to the geological time scale, the message from the Earth science community [70,71] that humans may alter the physical dynamics of Earth has led to a very committed debate putting geosciences at the center of the discussion about humans, society, and nature [72,73]. Hence, conscious of the physical processes, the sociopolitical drivers, and the cultural-ethical features of the human niche, contemporary thinkers and leaders did seek alternatives to the current hegemonic practices. Various branding has been proposed to label them; for example, "Mother Earth/Gaia", "Anthropocene", or "Ecomodernism" [48,74–76].

Within the development of Western societies over several centuries, science evolved from an undertaking of a minuscule elite to a massive societal endeavor [77–79]. Subsequently, the ethics of science gained relevance as a deontology and a societal feature [80,81]. Within that societal context, the explicit ethics of geosciences is a relatively recent field [82], though with multiple precursors, such as Leopold [83]. Subsequently, the sociohistorical conditions matured, first to acknowledge anthropogenic global change, and second to consider ethical frameworks (like geoethics), which also consider the abiotic compartment of Earth as a distinct feature of the human condition [84,85].

## 2.3. Affective Connotations of Geoethics

The wording that is used to describe what Geoethics is about also has conveying cognitive and affective meanings. Both are needed for meaningful human practices [27,28,86–88].

For example, the central tenet of Geoethics is the virtuous and responsible individual (human agent) who pursues a practice that is geosciences-knowledge-based, just, equitable, inclusive, participatory, and ecologically oriented [8]. That is an aspirational description of what ought to be done. Likewise, the central paradigms of Geoethics are reformulated in the geoethical promise [89] or the Cape Town Statement on Geoethics [90], which also uses phrasing that triggers affective associations. Correspondingly, Geoethics considers geo-heritage, geo-diversity, and geo-conservation to be critical cultural values. Three highly valued terms get "geo-tagged". More generally, geoethics seeks a human practice that valorizes geoscience expertise for beneficial societal use while protecting the "common heritage of mankind" for future generations. Hence, as the examples show, going beyond cognitive insights, Geoethics offers an affect-laden meaning. That is effective, as the irritation of some shows:

> . . . the whole thing [geoethics] consists of an idealist-based approach with the only difference with respect to religious idealism being that geoethics is, supposedly, secular . . . According to such approach, the world is idealised as a sum of individual atoms deprived of any social character and of any capability of social and collective actions but, rather, their possibility to act upon society is restricted only to their actions as individuals. It is easy to see that . . . any possibility to change the state of thing is limited to a sort of secular version of the Christian charity, . . . there is obviously not even a single word about the structural determinations upon individuals in the particular form of social organisation where they live. [91]

### 2.4. The Term Geoethics in the Scientific Literature

On the one hand, publications about the work of international geoscience bodies used the term geoethics for the first time less than twenty years ago [92]. It appears in contexts that describe geosciences as a service to society and implicitly consider ethics [57,93]. Before that time, the term geoethics was used at scientific conferences and in reports [94–96]. In these cases, the term has a meaning that relates to the current definition of geoethics [13], although with variations [97].

On the other hand, the scientific literature of the last three decades offers various uses of the term geoethics. Each of them matches the term, although they address planet Earth, Nature, and World by taking different perspectives. They refer to political sciences [98] or various aspects of geography [99–103]. Likewise, notions are used in geology (e.g., "geo-logic") to alternatively label thinking that strongly overlaps with Geoethics [104,105], and notions like "geographical ethics" are used [106], which may be easily compared to geo-ethics. Making matters further complex, schools of ethical thinking are found in other sciences of the environment, for example, in forestry [107] that have much in common with geoethical thinking. For example, they share claims as to who were their precursors, such as Aldo Leopold [83], and associate professional ethics as a prominent feature of their concepts. Finally, within geosciences, two distinguishable school of thinking exist that both use the label "geoethics". A dedicated school of thinking is found [108,109] that has its roots in problems related to mineral extraction as they were analyzed in the 1990s, mainly by scholars from Eastern Europe [110]. This school of thinking applies environmental ethics in geosciences and planetary sciences [82,111] (p. 13). The peculiarity of that school of thinking within environmental ethics is the study subject, namely, the abiotic process on Earth and other planets.

Hence, reviewing scientific literature shows that the use of the term "geoethics" is not univocal within the sciences and does not relate to a single concept or definition, not even within the geosciences or natural sciences. Nevertheless, in the broader geography-geoscience socio-professional context, the notion of geoethics is meaningful, although not unequivocal. Thus, following in the footprints of "sustainability", branding "geoethics" sounds like a promising idea.

### 2.5. Using the Term Geoethics in the Public Sphere

Promoting the term geoethics outside of the geosciences (or without tight coupling to professional ethics of geosciences) faces multiple challenges.

The term geoethics carries a "spontaneous" meaning. "Ethics" is understood to be about moral obligations toward human and non-human living beings, and "geo" relates to the world around us; hence, geoethics. Therefore, the apparent spontaneous understanding is about doing the right thing on Earth, be it to protect the environment or to promote sustainability. Subsequently, the term geoethics appears as a synonym for environmental ethics, or sustainability ethics, or a subtheme of both. Therefore, it may be perceived to be either a redundant notion or a notion that is conveniently tuned to catch the attention of the public [112] because the term geoethics sounds appealing.

As well, the term geoethics may easily be associated with concepts like "Mother Earth", that is, thoughts about environmental justice [113] or a kind of "Gaian Ethic" [14]. In the latter case, even the term sounds like Geoethics, and the semantic roots of both words relate. Therefore, this essay intends to show why a geoethics brand deserves its own place within academia and up to the public sphere.

## 3. Discussion

At first instance, the term geoethics refers to ethical practices of (professional) geoscientists regarding human interactions with the abiotic compartments of the Earth system(s). Although that limited scope is valid, it is evident from the works in Geoethics that the biotic compartments of the Earth system(s) are included because of the interwoven-ness of biotic and abiotic processes. Nevertheless, the biological world is not at the forefront of contemporary works that refer to Geoethics. However, it has been the primary concern of the scholar William S. Lynn [26], who was the first to use the term geoethics systematically (in a thesis applying it to the moral standing of animals) and with a meaning that would apply widely:

> Geoethics merges the horizons of ethics and geo[sciences], or more accurately, the horizons of distinct readings of ethics and geo[science] ... This constellation includes situated knowledge, contextual interpretation, and society/nature relations. From this merging, I trace three implications. First, geoethics uses geo[science]'s insight into the importance of context to avoid the major pitfalls of analytic moral thought ... Second, ... geoethics emphasises a plurality of moral concepts situationally appropriate for a moral understanding of our world. Third, geoethics seeks a moral understanding that values the well-being of animals, humans and the rest of nature on our inextricably earth-bound and interconnected world ... "

(pp. 1–2; our replacement of 'graphy' by 'science')

Beyond such early uses of the term geoethics, a recent description of the state of the ethical implications, societal contexts, and professional obligations of the geosciences [9] shows that the application scope of geoscientists' works on ethical questions of their disciplines is likely much more comprehensive than only addressing professional geosciences [114,115]. That, in turn, would justify promoting the term geoethics as the branding of sustainable human practices that favor *"context-dependent in space and time and ethically sound choices ... [and] a strong awareness of the technical, environmental, economic, cultural and political limits existing in different socio-ecological contexts"* [13], a description that mirrors Lynn's views. To note further, little to nothing in this description is specific to geosciences; on the contrary, it is generic.

Beyond its scope within geosciences, the term geoethics also associates easily with ethical (professional) practices in other disciplines. Examples are earth sciences [116], engineering disciplines [117], or people-disciplines [118]. These disciplines share a concern about the interaction of the social, cultural, and political compartments of the Earth system(s) with the abiotic and biotic compartments of the Earth system(s). In that sense, geoethics may set a rationale for practices of citizens, namely a paradigm of a caretaker

who chooses *"between a path of care and a path of neglect"* [35] (p. 150) to act Earth-centrically and society-centrically [119].

Contrary to the apparent, the use of the term geoethics is ambiguous. Nevertheless, it is appealingly affect-laden and straightforward. Therefore, people may easily take it up because it makes sense to them and has some meaning. The analysis shows that multiple connotations may be related to the term; therefore, some confusion about the meaning of the notion of geoethics seems unavoidable, one may argue. However, a similar concern was (and is) present with the branding and meaning of the notion of sustainability [2,3], which also has diverse connotations. Nevertheless, the notion of sustainability is an accepted and successful branding today. Hence, an apparent difficulty for branding geoethics should not diminish the potential of the generic, core concept that is underneath it, squarely set as the *"reflection on the values that underpin appropriate behaviors and practices, wherever human activities interact with the Earth system"* [13].

It appears that the debate is closed as to whether Geoethics should apply only to professional geoscientists or any citizen, a debate which was opened by the authors [23]. The paradigm of geoethics is relevant to any citizen. However, the debates are still open as to how geoethical thinking, founded on another ethical basis rather than an actor-centric, virtue-ethic would shape its border zones with environmental ethics or sustainability ethics. Likewise, debates must be engaged as to how Geoethics, as philosophical thinking, relates to more general inquiries regarding human-geosphere intersections, inquiries that are not anchored in the geosciences only. Such inquiries were sketched already in 1947 by J.K. Wright [120] (p. 10) in his presidential address to the forty-third annual meeting of the American Association of Geographers when debating about geography:

> *My term is Geosophy, compounded from geo meaning "earth" and sophia meaning [wisdom] ... Geosophy, to repeat, is the study of geo[science] knowledge from any or all points of view ... Thus, it extends far beyond the core area of scientific geo[science] knowledge or of geo[science] knowledge as otherwise systematised by geo[scientists]. Taking into account the whole peripheral realm, it covers the geo[science] ideas, both true and false, of all manner of people—not only geo[scientists], but farmers and fishermen, business executives and poets, novelists and painters, Bedouins and [Khoikkois]—and for this reason it necessarily has to do in large degree with subjective conceptions. Indeed, even those parts of it that deal with geo[science] must reckon with human desires, motives, and prejudices, for unless I am mistaken, nowhere are geo[scientists] more likely to be influenced by the subjective than in their discussions of what geo[science] is and ought to be (our replacement of 'graphy' by 'science' and 'grapher' by 'scientist').*

In the face of a "defiant Earth" [35], J.K. Wright's concept of "geosophy" (a notion that was reused recently; see, for example, [121]) may subsume the concept that underpins Geoethics. However, and referring to the imperative of responsible action [122], both notions involve the three cultures of the 21st century [123], namely, natural sciences, social sciences, and the humanities. They encompass comprehensive inquiries into the natural features of the Earth system and the physical and cognitive artefacts of the technosphere [124], the latter including the ensemble of social, cultural, and political insights of people, as well as their affective perception of the Earth system.

## 4. Conclusions

Sustainability is a concept that has acquired recognition as a "brand" for sound dealings in the human niche. The term geoethics labels thinking that is situated at the intersection of sustainability ethics, environmental ethics, and the ethics of societal practices of citizens. The philosophical thinking labelled Geoethics that is found within geosciences is a subcase. Its relevance increases with the rise of anthropogenic global change because the geosciences provide expertise for the stewardship of abiotic features of the human niche and for its governance, which in turn would affect the biotic and cultural spheres.

The ethics of the societal practices of citizens include the professional ethics of geoscientists as an important subcase, because this variant of professional ethics is the foundation for the sound societal use of geosciences [125].

Geoethics within geosciences is a school of thinking that emerged with a discipline-specific application. Notwithstanding its origins, its design is not limited to a discipline-specific application. By coincidence, the chosen label, geoethics, is generic. Therefore, the term geoethics may serve, beyond other uses, as a systemic notion and a semiotic sign (brand) to promote the ethics of sound human practices within the abiotic, biotic, and cultural compartments of the (single) Earth system. What that implies in detail and in a specific circumstance may be debated, using the branding of sustainability as an example. The brand geoethics, understood also as a semiotic sign, would refer to Geoethics as an ethical framework, the widespread use of geoethical thinking and action as a societal practice, and the special attention to the societal use of geosciences expertise as paramount for sustainable stewardship of the human niche.

**Author Contributions:** Conceptualization, methodology, M.B.; writing and editing, M.B. and E.M. All authors have read and agreed to the published version of the manuscript.

**Funding:** This research received no external funding.

**Institutional Review Board Statement:** Not applicable.

**Informed Consent Statement:** Not applicable.

**Data Availability Statement:** Not applicable.

**Acknowledgments:** The authors like to thank their peers for fruitful discussions and three anonymous reviewers for advice that helped to balance the paper.

**Conflicts of Interest:** The authors declare no conflict of interest.

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
