# Peer review of "Geoethics, a Branding for Sustainable Practices"

_sustainability, doi:10.3390/su13020895_

Round 1
Reviewer 1 Report
I want to congratulate to authors doing this philosophical paper that lights the way of Geoethics.
The title clearly describes the article. The abstract reflects the content of the ms. Finally, the paper describes that author hopes to achieve accurately.
And regarding everything else I wish to express a rhetorical question: Is Geoethics a science or a speculation?
Author Response
- see attached file -

Reviewer 2 Report
The reviewed text is dealing with interesting and timely topic. Although the text is well structured, it is actually very hard to read and core of the message is not clearly stated. Please, revise the text keeping in mind that it should be understandable to general readership and even to those who might not have that much experience in geoethics and geosciences. At the moment the text is more a collection of impressions on geoethics rather than discussion actively comparing different approaches and ideas. But what should a general reader really take from your text? What is to be learned from your text also by those who are not involved in geoethics? If this is to be a theoretical article, it should be revised in a more systematic manner and involve critical discussion. Doscussion should also involve assesment and comparison. In the abstract you mention that you aim to discuss how geoethics has been applied and may be used. Could you provide and discuss practical examples? Also, considering geoethics as a movement, is it institutionalized? Where does it come from? Within which countries or parts of the world has it developed? The text is sometimes sketchy, including just short statements that are not always explained in a sufficient way. For example, on the line 288 you mention the concept of geosophy by J. K.Wright, but never explain the original meaning of this concept and its significance for geography. Why you mention it, without further discussing it? Similar problems or lack of explanation are to be found through the whole text. I also am not sure, if I understand why you use the words brand and branding and what you mean by these in the context of your article. The title of the section 2 "Materials, methods and results" does not fit its contents. Rework the title, so it characterizes the text under it. Please, mention clearly the aim of your text in the abstract.
Author Response
- see attached file -

Reviewer 3 Report
A rather "borderline" work that tries, moreover, to clarify some concepts, terms, use, etc. in a contextual framework of sustainability
Author Response
- see attached file -

Round 2
Reviewer 2 Report
I would like to thank the authors for considering the comments. The changes authors made to the text, improved it and the article may be accepted for publication.
Just a few words to add for clarification. As to the term "brand", my comments were based on general knowledge of the term and its usage in various contexts. Although I understand, why you use the term "brand", in my oponion, for your purpose it might better to consider the concept of "paradigm" instead of brand, especially as you refer to geoethics as "a school of thought that uses established philosophical concepts". Also having geographical background, it might be that I have just read your text from a different perspective than the other reviewers.
This manuscript is a resubmission of an earlier submission. The following is a list of the peer review reports and author responses from that submission.